# RLBFF: Binary Flexible Feedback to bridge between Human Feedback & Verifiable Rewards

**Zhilin Wang, Jiaqi Zeng, Olivier Delalleau, Ellie Evans, Daniel Egert,**
**Hoo-Chang Shin, Felipe Soares, Yi Dong, Oleksii Kuchaiev**
NVIDIA
`{zhilinw, jiaqiz}@nvidia.com`

## Abstract

Reinforcement Learning with Human Feedback (RLHF) and Reinforcement Learning with Verifiable Rewards (RLVR) are the main RL paradigms used in LLM post-training, each offering distinct advantages. However, RLHF struggles with interpretability and reward hacking because it relies on human judgments that usually lack explicit criteria, whereas RLVR is limited in scope by its focus on correctness-based verifiers. We propose Reinforcement Learning with Binary Flexible Feedback (RLBFF), which combines the versatility of human-driven preferences with the precision of rule-based verification, enabling reward models to capture nuanced aspects of response quality beyond mere correctness. RLBFF extracts principles that can be answered in a binary fashion (e.g. accuracy of information: "yes", or code readability: "no") from natural language feedback. Such principles can then be used to ground Reward Model training as an entailment task (response satisfies or does not satisfy an arbitrary principle). We show that Reward Models trained in this manner can outperform Bradley-Terry models when matched for data and achieve top performance on RM-Bench (86.2%) and JudgeBench (81.4%, #1 on leaderboard as of September 24, 2025). Additionally, users can specify principles of interest at inference time to customize the focus of our reward models, in contrast to Bradley-Terry models. Finally, we present a fully open source recipe (including data) to align Qwen3-32B using RLBFF and our Reward Model, to match or exceed the performance of o3-mini and DeepSeek R1 on general alignment benchmarks of MT-Bench, WildBench, and Arena Hard v2 (at $< 5\%$ of the inference cost).

🤗 **Models:** huggingface.co/collections/nvidia/reward-models-10-2025

Table 1: Comparison of Human Feedback with Verifiable Rewards for Reinforcement Learning, alongside our proposed Binary Flexible Feedback serving as a bridge between the two. See the Introduction for rationales behind the classification into good (✓) and poor (✗) on various aspects.

| | Wide Coverage | Interpretability | Precision | Recall |
|---|:---:|:---:|:---:|:---:|
| RLHF - Human Feedback (Bai et al., 2022) | ✓ | ✗ | ✗ | ✓ |
| RLVR - Verifiable Rewards (Lambert et al., 2025) | ✗ | ✓ | ✓ | ✗ |
| RLBFF - Binary Flexible Feedback (This work) | ✓ | ✓ | ✓ | ✓ |

## 1 Introduction

Reinforcement Learning with Human Feedback - RLHF (Ouyang et al., 2022; Bai et al., 2022) and Reinforcement Learning with Verifiable Rewards - RLVR (Lambert et al., 2025; DeepSeek-AI et al., 2025) are two popular paradigms currently used for training Large Language Models (LLMs) with Reinforcement Learning (RL). Recent open-weight general-purpose LLMs are trained with both RLHF and RLVR (Yang et al., 2025a; Team et al., 2025; Gemma et al., 2025) because of their complementing strengths. While these techniques have distinct advantages, we propose Reinforcement Learning with Binary Flexible Feedback - RLBFF - that can bridge both techniques to combine their respective benefits.

**Formulation** To formulate RLBFF, we first adopt RLVR's approach of using binary rewards: given a prompt and response, a verifier will either give the response full reward if the response is correct based on the verifier or no reward otherwise. We notice a parallel between this and the rewards used in KTO (Ethayarajh et al., 2024) where a general-domain response can be marked as either good or bad. A critical difference however lies in that this binary signal in KTO may be tied to various aspects of response quality (e.g., correctness, helpfulness, coherence, etc.) that are not explicit in the data. Owing to this difference, we find that representing the principle(s) for which a response is marked as good/bad can be useful - RLVR only uses the principle of correctness while KTO does not explicitly define the principle(s), meaning that the judgment is based on an *unknown* combination of principles. Our proposed format is hence: given a prompt, response and principle, indicate whether the response fulfills the principle.

**Motivation** To arrive at this formulation, we made a few design choices:

1. **Why principles?** In various situations, the reasons why humans like/dislike a response can be due to different principles. For instance, on some sub-reddits, the most highly up-voted comment is not necessarily the most helpful or correct but the most hilarious. Conversely, the most up-voted response on StackExchange-Math tends to be the most correct. Learning preferences without *explicitly* considering the principle behind judgments can make training less effective, since the optimization objective becomes less clear (OpenAI, 2025).

2. **Why single response instead of response pair?** While response pairs to a common prompt is usual for RLHF, we find that this is unnatural for most settings where people provide textual feedback online. For instance, when humans review a restaurant or a product, it's mostly focused on various aspects of itself. While some implicit comparisons can occur (e.g. this is the best Thai food place in town), explicitly comparing restaurant A to restaurant B is much rarer (Keller & Kostromitina, 2020). Response pairs are also prone to position bias (Zheng et al., 2023).

3. **Why binary (instead of Likert)?** Bai et al. (2022) and Ouyang et al. (2022) found that Likert scoring (e.g. 5-point scale) for evaluating LLM response can be hard to calibrate across different people, since they can have different expectations on what a score:3 response is vs. score:4. At the principle level, this is even harder because for instance a very concise response can be harder to separate from a concise response: binarizing the possible options (e.g. concise vs. not concise) reduces such annotation disparities.

We compare this formulation of Binary Flexible Feedback with Human Feedback and Verifiable Rewards, as summarized in in Tab. 1.

**Wide Coverage** Human Feedback can be used for a wide range of tasks that users ask of LLMs like ChatGPT (Ouyang et al., 2022). Conversely, Verifiable Rewards are typically reserved for problems with easy-to-verify correctness, including math problems with a single correct answer (Lambert et al., 2025), competitive coding problems similar to LeetCode (DeepSeek-AI et al., 2025) and precise instruction following to check if for instance, a certain word appears n times correctly (Pyatkin et al., 2025). Binary Flexible Feedback inherits the versatility of Human Feedback since principles can include any aspect that humans value (beyond only correctness).

**Interpretability** Given that Human Feedback is typically in form of response A is better than response B (Bai et al., 2022; Wang et al., 2025a), a trained Bradley-Terry model produces scores that are not globally calibrated across prompts. This means that the score of each response (e.g. -14.5) can only be used and interpreted in the context of the scores of other responses to the same prompt. In addition, such a model usually operates as a black-box, i.e., it provides no explanation as to why a response received a given score. On the other hand, Verifiable Rewards are easily interpretable (i.e. either Yes or No in terms of correctness), and so is Binary Flexible Feedback.

**Precision and Recall** Reward Models trained with Human Feedback are known to be affected by low precision - also commonly known as Reward Hacking (Weng, 2024). Specifically, this happens when the model allocates high reward to a response due to features that are not widely accepted to support response quality, including matching user's beliefs (Sharma et al., 2023) or higher response length (Dubois et al., 2025). Verifiers, on the other hand, suffer from the opposite problem of low

> The response is mostly helpful. It resolves the issue directly, provides the corrected full code, and follows the user's requirements - *follows the user's requirements: yes*. It correctly interprets the real intent of the user and fixes the correct line. However, it doesn't have any inline comments - *includes inline comments: no*, especially where the update is done. It could be better if the code contained inline comments to line 5, where '$<=$' is replaced with '$<$'.

Figure 1: Example of Binary Flexible Feedback in Natural Language. *Text in italics* are generated principles and their fulfillment; Highlighted spans are evidence that supports the principle identification with green highlight indicating fulfillment while yellow highlights indicates non-fulfillment.

recall because they may fail to recognize correct answers, which are equivalent to the reference correct answer (e.g., 3 hours vs. 180 minutes *or* $0.5\pi$ vs. $90°$ in geometry) (Huang et al., 2025). This means that they can overly penalize correct answers. Binary Flexible Feedback reduces reward hacking by identifying a specific principle/feature for modeling, and reduces failure to recognize equivalent correct answers by training on top of LLMs that have been pretrained to recognize such equivalence between equally correct answers. We elaborate upon its theoretical basis in App. A.

**Training Reward Models using Binary Flexible Feedback**   To empirically test the effectiveness of Binary Flexible Feedback, we attempt to find such data without success. However, we notice that the open-source dataset HelpSteer3-Feedback (NVIDIA, 2025a) can be easily converted into such format. We propose a method for converting feedback in natural language into a set of Binary Flexible feedback with an example shown in Fig. 1. We then use this data to train top-performing Reward Models on RM-Bench (Liu et al., 2025a), JudgeBench (Tan et al., 2025) and PrincipleBench, a new human-annotated evaluation benchmark that we introduce to measure the effectiveness of Reward Models in adhering to specific principles. Finally, we show performance of model alignment with RLBFF, by leveraging the same principles obtained from the HelpSteer3 human feedback and our trained reward model.

**Main Contributions**

1. Reinforcement Learning with Binary Flexible Feedback, which combines the benefits of RLHF and RLVR. Reward Models trained on this technique have top performance on JudgeBench (81.4%, #1 on leaderboard as of 24 Sept 2025), RM-Bench and PrincipleBench.

2. PrincipleBench, a benchmark to measure reward models' ability to follow specific principles when assigning reward, a feature not seen in prior public Reward Modeling benchmarks.

3. Fully open-source formula (including data) to align Qwen3-32B with RLBFF to match or exceed the performance of proprietary models like o3-mini and DeepSeek R1 on MT-Bench, WildBench and Arena Hard v2 (at $< 5\%$ of the inference cost).

## 2 RELATED WORK

**Binary Flexible Feedback for Safety and Math**   Previous work has explored binary feedback in narrow domains. For safety, Mu et al. (2024) uses rule-based checks that ask an off-the-shelf LLM whether a response complies with specific policies (e.g., includes a brief apology and refusal). For math, Zhang et al. (2024) trains generative verifiers to judge whether an answer is correct. These approaches operate over a small, fixed inventory of ($\approx 10$) principles. In contrast, our formulation scales to 1,000+ fine-grained principles spanning general, STEM, code, and multilingual domains, substantially broadening coverage while retaining binary evaluability.

**Generative Rewards Models with Self-Generated Criteria**   DeepSeek-GRM (Liu et al., 2025b) and RM-R1 (Chen et al., 2025) learn to predict preference rankings from open-source datasets by first synthesizing rubric criteria and then grading responses against those criteria. While these self-generated rubrics provide some grounding for preference judgments, they are not user-controllable: at evaluation time, users cannot specify or swap in custom principles.

**Principle-Following Generative Reward Models** RewardAnything (Yu et al., 2025) manually curates 200 criteria spanning Content, Structure, Logic, Tone, and Style, then uses an LLM ensemble (Claude-3.7 Sonnet, GPT-4.1, DeepSeek-V3, Gemini 2.5 Pro) to assign Likert-5 labels indicating whether each response satisfies each criterion. R3 (Anugraha et al., 2025) bootstraps rubrics by consolidating task-specific annotation criteria from 10+ datasets (e.g., UltraFeedback) into pseudo-rubrics, leveraging the datasets' ground-truth labels for supervision. LMUnit (Saad-Falcon et al., 2024) mixes both strategies: it incorporates HelpSteer2 (Wang et al., 2024) annotation criteria plus 10 coarse, hand-crafted rubrics, and augments them with synthetically generated fine-grained principles from an in-house set and Prometheus dataset (Kim et al., 2024). In contrast, our approach operates over >1,000 distinct, fine-grained principles derived directly from human-written feedback rather than synthetic generation and yielding much broader coverage while preserving a straightforward, binary (yes/no) scoring scheme.

## 3 TRAINING DATA

**Downloading HelpSteer3-Feedback** from HuggingFace (NVIDIA, 2025a). This dataset contains 40,821 samples with each sample consisting of one prompt (possibly multi-turn), two responses and up to three human-written paragraph-length (2–10 sentences or 50–250 words) textual feedback per response. This feedback was written in English by 7000+ human annotators across 80+ regions for samples in General, STEM, Code and Multilingual domains (Wang et al., 2025b).

**Extracting Principles and Fulfillment** We define a principle as an axis upon which a response can be evaluated in a binary fashion. Rather than pre-defining a fixed set of principles, we identify principles relevant to each response. We use DeepSeek V3-0324 – the strongest open-weight, non-reasoning model at the time of our experiments – since this task does not require advanced reasoning capabilities, thus substantially reducing the compute required for principle extraction. We prompt the model to generate greedily in a JSON output format, with additional fields for supporting text spans and yes/no/partially for whether the text span supports the principle. The format was followed in 99.9% of generations and the remaining were excluded. We generate principles using a zero-shot prompt template (see App. B) as early few-shot experiments heavily biased principle distributions.

**Filtering Principles Unsupported by Feedback** Our initial spot-checks suggest that generating principles without supporting text spans frequently leads to hallucinations. Therefore, our final prompt (see App. B) involves asking the model to cite evidence of a supporting text span from the human-written feedback, prior to answering the satisfaction of the principle. In such setting, the model could still generate a non-existent supporting text span or slightly paraphrase the original text span. We reject extractions where the cited span does not plausibly come from the feedback using the *Rapid-Fuzz* string matching library (Bachmann, 2025) with `rapidfuzz.partial_ratio(feedback, text_span) > 60`, removing 2.2% of principles. Such an evidence-citation mechanism minimizes LLM hallucinations relative to synthetic principle generation approaches that are not grounded in human-written feedback. In addition, we exclude all 'helpfulness' principles (since helpfulness here refers to the *global* quality assessment of responses – rather than being based on a particular principle). Prior to exclusion, it accounted for 4.5% of raw principles due to an artifact within HelpSteer3-Feedback that all human-written feedback starts with 'The response is . . . helpful'.

**Removing Partially Fulfilled Principles** While we recognize that some principles can be partially fulfilled, natural language does not offer a clean way of determining whether partial means 10%, 25%, 50%, 75% or 90%. Similarly, using extent-markers in natural language such as slightly or moderately also does not offer precision, as different annotators might have different understandings of what each word means. We thus remove the principles marked as "partially" fulfilled (only 13.8%, suggesting that a binary value is suitable for most principles). Among the remaining principles, 35.4% are no and 64.6% are yes - indicating only a slight imbalance, the effect of which we study in App. H.

**Obtaining High-Precision Consensus Principles** Inspired by HelpSteer2 Wang et al. (2024), we find that identifying consensus across annotators can be important to reduce outlier annotations. This is because the perspective of a single annotator can be subjective, which does not necessarily reflect the consensus. However, unlike HelpSteer2 where aspect ratings have numerical values that can be easily used to calculate consensus, principles are in free text. Different annotators might word

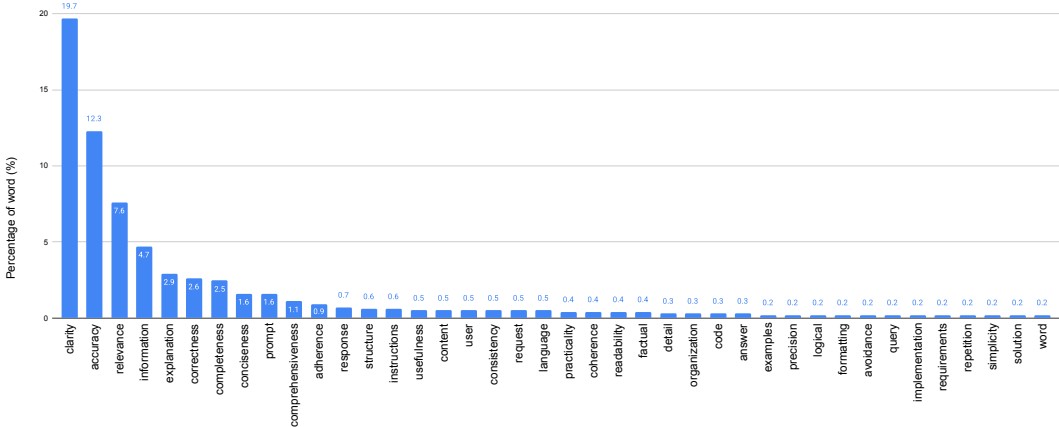

Figure 2: 40 most frequent words in principles, excluding stop-words. Clarity, accuracy and relevance are most common, followed by a long tail including comprehensiveness, readability and precision.

similar principles using different terms (e.g. correctness vs accuracy vs acccuracy of information). To solve this challenge, we use Qwen-3-8B Embedding (Zhang et al., 2025), which was the highest scoring model on an embedding benchmark - MTEB (Muennighoff et al., 2023) when we tackled this problem. We only keep principles that have at least one principle from each of all other annotators with cosine similarity > 0.8 (after spot-checking 0.7/0.8/0.9 and finding that 0.8 leads to matching synonyms without requiring word-for-word matching). This step was by far the most stringent filter, as we only retained around 100 thousand principles (across 3 annotators, or 33k principles with 'unique' meanings) out of an initial 1.2 million. Our filtering is intentionally high-precision and low-recall, with the expectation that some correct principles will be filtered out and retaining only 1.27 principles per feedback on average (std of 0.543). We believe that such a tradeoff is helpful, since we have many principles to start with and this guards against training on mis-specified criteria.

**Human Verification** To ensure that extracted principles are faithful to the natural language feedback, we conducted a small human verification exercise with 3 volunteer human annotators, who each took less than 1 hour. Each annotator independently annotated 126 random samples, with each sample containing a natural language feedback and the corresponding extracted principle. Specifically, they were asked if the principle as well as the yes/no answer adhered to the natural language feedback. We found the inter-rater agreement to reach 0.447 Fleiss' $\kappa$ (moderate agreement). The majority annotator answer for each principle agreed with the extracted principle in 88.9% of cases, suggesting the robustness of principle extraction.

**Principle Distribution** Out of the 33k principles, there are 1,414 unique principles. Most are single-word (55.0%; e.g., *clarity*), followed by three-word (35.2%); the rest are two-word (5.2%), four-word (4.1%), and five-word (0.5%), with <0.1% in 6–9 words. Among principles longer than 2 words, most are conjoined by "of" (24.9%) like *clarity of guidance*, "and" (8.6%) or "to" (5.3%). There also exist negated principles such as *avoidance of repetition*. Fig. 2 shows the most frequent words represented in principles, indicating a diverse set of principles along many semantic axes.

## 4 REWARD MODELING

### 4.1 EVALUATION

**RM-Bench and JudgeBench** Following HelpSteer3-Preference (Wang et al., 2025c), we use two popular benchmarks RM-Bench (Liu et al., 2024) and JudgeBench (Tan et al., 2025) for evaluation. Evaluation numbers for the baseline models are from the original papers, except in situations where THU-KEG (2025) found a discrepancy between reported numbers due to an inaccurate formula used for RM-Bench [1] or when JudgeBench numbers were unavailable.

---

[1]Macro-avg across domains vs. sample-avg inaccurately reported by Yu et al. (2025) Details in App. D

**PrincipleBench** As RM-Bench and JudgeBench are predominantly focused on the correctness of responses, we believe it is important to evaluate Reward Models' ability to follow principles on other aspects such as response clarity. There is no such open-source benchmark available[2] but we saw an opportunity to create one. Specifically, we noticed that HelpSteer3-Preference (Wang et al., 2025c) collected (but did not release) human annotations on several binary principles: 'contains incorrect information', 'key information is missing', 'misses one or more specific prompt requirement(s)', 'contains repetitions', 'irrelevant information', 'contains awkward phrasing / formatting issue' and 'response does not follow prompt language'. We were successful in obtaining additional human annotations in the validation subset (never used for training) from these authors after reaching out.

We create PrincipleBench with re-worded versions of this data (App. B). Since each binary principle has been human-labeled by 3 annotators, we only keep principles for each sample that all three annotators agree on, to avoid accidentally including labels that contain errors. To derive a pair of chosen-rejected responses, we also only keep samples for which one response fulfills a principle while the other does not. This benchmark contains 487 samples across General, STEM, Code (with 14 programming languages) and Multilingual (with 13 natural languages) in total, similar in size to JudgeBench. In addition, PrincipleBench contains not only single-turn prompts but also context up to 10 prior turns where only the last assistant turn is considered during evaluation. Following JudgeBench, we consider a pairwise GenRM to get a task correct only when it correctly identifies the chosen response both when it is presented before and after the rejected one, in order to minimize position bias. We report sample-level micro-average as the Overall score, inspired by JudgeBench.

## 4.2 BASELINES

**Scalar Reward Models** We use a Bradley-Terry Reward Model trained with similar preference data (i.e. same prompts and responses) from Wang et al. (2025c) as a baseline. In addition, we want to explore how the Binary Flexible Feedback would work with a single fixed principle. Therefore, we used HelpSteer3-Feedback to formulate the principle of 'helpfulness' based on whether responses were marked by all three annotators as perfectly/mostly helpful (labeled Yes), or slightly/not helpful (labeled No). Prompt Templates and training details are in App. B and C respectively. In addition, we also use off-the-shelf Reward Models with similar model size as baselines, including Llama-3.3-Nemotron-70B-Reward and Llama-3.1-Nemotron-70B-Reward.

**Generative Reward Models** We use Llama-3.3-Nemotron-Super-49B-GenRM, trained with similar data (i.e. same prompts and responses) from Wang et al. (2025c) as a baseline. In addition, we use RewardAnything-8B-v1 (Yu et al., 2025), RM-R1-DeepSeek-Distilled-Qwen-32B (Chen et al., 2025) and R3-QWEN3-14B-LORA-4K (Anugraha et al., 2025) since they are the strongest models from each of these related work. We ran inference on each model following the recommended hyper-parameters and prompt template on the HuggingFace Model page and papers. We also planned to run LMUnit (Saad-Falcon et al., 2024) models but the HF model pages (ContextualAI, 2025a;b) were placeholders (i.e. empty) and we did not have sufficient information to infer with these models.

## 4.3 TRAINING

**Scalar Reward Models** We train Scalar RMs that predict reward scores with a single generated token equivalent of compute. Following Wang et al. (2025c), we start with Llama-3.3-70B-Instruct as the base model[3] and train it to predict either Yes or No, given a prompt, response and principle. At evaluation time, we calculate the reward as the difference between the log-prob of Yes and the log-prob of No, conditioned upon a prompt, response and principle that best describes the evaluation sub-category. This evaluation approach is inspired by Zhang et al. (2024) and Kadavath et al. (2022), but we added support for flexible principles while prior works only supported one fixed principle. The advantage of this method lies in extreme efficiency – requiring only 1 generated token of compute during inference, whilst also providing an estimation of the confidence in fulfilling a principle, beyond getting only 'Yes' or 'No'. Prompt templates and training details are in App. B and C respectively. Since this model can be flexibly used with any principle, we refer to it as Flexible Principles hereafter.

---

[2]Yu et al. (2025) proposed RABench, which is similar but is LLM-labelled and not publicly available yet.

[3]An initial experiment with a smaller Llama-3.1-8B-Instruct as the base model can be found in App. G

**Generative Reward Models** We follow Wang et al. (2025c) to train Generative RMs (GenRMs) with reinforcement learning using the GRPO (Group Relative Policy Optimization) algorithm (Shao et al., 2024). Starting with Qwen3-32B (Yang et al., 2025b), we provide the model with a conversation history and a principle, and train it to first reason through the task and principle, then give a final judgment Yes or No. Similar to scalar reward models, we define the reward as the difference between the log-prob of Yes and the log-prob of No, and use it during both training and evaluation. Equipped with reasoning capabilities, GenRMs are designed for more complex tasks that require step-by-step reasoning before assigning rewards. On the other hand, the step-by-step reasoning also means that GenRMs are substantially ($\approx 2$ orders of magnitude) slower and more compute-intensive during both training and inference. Given this constraint, we only train a GenRM with the best data configuration found after training various Scalar RMs. Training details and prompt templates are in App. B and C.

## 4.4 RESULTS

Table 2: Performance of Reward Models on RM-Bench and JudgeBench. Higher is better.

| Model | RM-Bench | | | | | | | | JudgeBench | | | | |
| | Chat | Math | Code | Safety | Easy | Normal | Hard | **Overall** | Knowl. | Reason. | Math | Coding | **Overall** |
|---|---|---|---|---|---|---|---|---|---|---|---|---|---|
| Scalar RMs (<0.1 second/task) | | | | | | | | | | | | | |
| *Ours* | | | | | | | | | | | | | |
| Flexible Principles ScalarRM | 85.3 | 81.9 | 70.4 | 96.9 | 85.5 | 84.9 | 80.5 | **83.6** | 74.0 | 74.5 | 82.1 | 81.0 | **76.3** |
| Bradley-Terry | 73.6 | 82.7 | 66.1 | 91.4 | 89.2 | 82.0 | 64.2 | 78.5 | 63.0 | 69.4 | 82.1 | 71.4 | 68.9 |
| *External Baselines* | | | | | | | | | | | | | |
| Llama-3.3-Nemotron-70B-Reward | 75.4 | 84.5 | 69.3 | 90.4 | 92.1 | 84.1 | 63.5 | 79.9 | 70.8 | 76.5 | 82.1 | 66.7 | 73.7 |
| Llama-3.1-Nemotron-70B-Reward | 70.7 | 64.3 | 57.4 | 90.3 | 92.5 | 76.4 | 43.1 | 70.7 | 62.3 | 72.5 | 76.8 | 57.1 | 66.9 |
| Generative RMs (>10 seconds/task) | | | | | | | | | | | | | |
| *Ours* | | | | | | | | | | | | | |
| Flexible Principles GenRM | 80.4 | 92.0 | 77.0 | 95.5 | 88.9 | 86.4 | 83.4 | **86.2** | 74.6 | 85.7 | 85.7 | 90.5 | **81.4** |
| *External Baselines* | | | | | | | | | | | | | |
| Llama-3.3-Nemotron-Super-49B-GenRM | 73.7 | 91.4 | 75.0 | 90.6 | 91.2 | 85.7 | 71.2 | 82.7 | 71.4 | 73.5 | 87.5 | 76.2 | 75.1 |
| RewardAnything-8B-v1 | 76.7 | 90.3 | 75.2 | 90.2 | 85.6 | 82.2 | 81.5 | 83.1 | 61.0 | 57.1 | 73.2 | 66.7 | 62.6 |
| RM-R1-DeepSeek-Distilled-Qwen-32B | 74.2 | 91.8 | 74.1 | 95.4 | 89.5 | 85.4 | 76.7 | 83.9 | 56.5 | 66.3 | 85.7 | 73.8 | 66.0 |
| R3-QWEN3-14B-LORA-4K | 76.5 | 92.4 | 78.7 | 91.9 | 91.4 | 86.2 | 77.1 | 84.9 | 50.0 | 64.3 | 76.8 | 71.4 | 60.9 |

**Flexible Principles are the top performing model across Scalar and Generative RMs** As shown in Tab. 2 and 3, the Flexible Principles model is top among Scalar RMs across RM-Bench (83.6), JudgeBench (76.3) and PrincipleBench (91.6). As Generative RM experiments are computationally intensive (see App. C), we only apply it to our best-performing ScalarRM recipe (Flexible Principles). Our Flexible Principles GenRM improves upon the Flexible Principles ScalarRM to achieve a further improvement on RM-Bench (86.2) and JudgeBench (81.4) - which is higher than all baseline GenRMs as well as the top of JudgeBench leaderboard (80.9) (Tan et al., 2025) as of 24 Sep 2025.

**Poor Performance of Baseline GenRMs on JudgeBench** We notice that while many Baseline GenRMs do well on RM-Bench, their performance were poor on JudgeBench, often underperforming Scalar RMs. For instance, Tab. 2 shows that RewardAnything-8B-v1 achieved only 62.6 on JudgeBench, which is below the lowest performing Scalar RM (Llama-3.1-Nemotron-70B-Reward at 66.9) despite performing much better on RM-Bench. To better understand why this is so, we attempt an experiment where instead of requiring both orders (chosen-first *and* reject-first) to get the answer right, we separately require only i. chosen-first, or ii. rejected-first. We find that the model performs well in chosen-first order at 77.1 which drops to 65.1 for rejected-first order, and further to 62.6 when both orders need to agree. This suggests substantial position bias for pairwise (or n-wise) Generative RMs, while our design of rating responses individually averts this bias, leading to a SOTA performance on JudgeBench.

**Flexible Principles is the first Scalar RM to enable grounding by user-specified principles** While RewardAnything-8B-v1 and R3-QWEN3-14B-LORA-4K enabled grounding by user-specified principles previously, both of them are reasoning GenRMs that requires thousands of generated tokens per sample at inference time (while our Scalar RM only requires 1 generated token per sample). This means that our Flexible Principles Scalar RM can finish each task in <0.1 second while GenRMs take >100x longer - while achieving similar or better benchmark performance across RM-Bench, JudgeBench and PrincipleBench. This makes the Flexible Principles Scalar RM ideal for latency-sensitive settings that require users to set custom principles for scoring.

Table 3: Performance of Reward Models on PrincipleBench. Higher is better for each category.

| | Clar-ity | Accu-racy | Relev-ance | No Rep-etition | Aspects Lang. Alignm. | Essen. Info. | Requir. Complete | Macro-Average | General Chat | STEM Sci./Math | Code Prog. | Multi-lingual | Domains Macro-Average | Overall Micro-Average |
|---|---|---|---|---|---|---|---|---|---|---|---|---|---|---|
| **Scalar RMs (<0.1 second/task)** | | | | | | | | | | | | | | |
| *Ours* | | | | | | | | | | | | | | |
| Flexible Principles ScalarRM | 90.6 | 89.4 | 94.9 | 100 | 87.5 | 92.1 | 90.0 | **92.1** | 89.0 | 89.6 | 94.8 | 93.7 | **91.8** | **91.6** |
| Bradley-Terry | 84.4 | 91.5 | 89.8 | 89.7 | 83.3 | 88.8 | 90.7 | 88.3 | 90.4 | 87.5 | 88.2 | 90.5 | 89.2 | 89.5 |
| *External Baselines* | | | | | | | | | | | | | | |
| Llama-3.3-Nemotron-70B-Reward | 87.5 | 90.4 | 93.2 | 89.7 | 66.7 | 89.9 | 92.0 | 87.1 | 92.8 | 87.5 | 89.6 | 84.2 | 88.5 | 89.7 |
| Llama-3.1-Nemotron-70B-Reward | 93.8 | 81.9 | 91.5 | 89.7 | 66.7 | 87.6 | 90.7 | 86.0 | 90.0 | 85.4 | 88.9 | 81.1 | 86.3 | 87.5 |
| **Generative RMs (>10 seconds/task)** | | | | | | | | | | | | | | |
| *Ours* | | | | | | | | | | | | | | |
| Flexible Principles GenRM | 84.4 | 84.0 | 79.7 | 82.1 | 100 | 82.0 | 84.0 | **85.2** | 81.3 | 70.8 | 87.4 | 90.5 | **82.5** | **83.8** |
| *External Baselines* | | | | | | | | | | | | | | |
| Llama-3.3-Nemotron-Super-49B-GenRM | 81.2 | 80.9 | 81.4 | 89.7 | 62.5 | 85.4 | 82.7 | 80.5 | 83.3 | 66.7 | 88.1 | 78.9 | 79.3 | 82.1 |
| RewardAnything-8B-v1 | 78.1 | 61.7 | 62.7 | 82.1 | 87.5 | 80.9 | 75.3 | 75.5 | 68.9 | 66.7 | 76.3 | 83.2 | 73.8 | 73.5 |
| RM-R1-DeepSeek-Distilled-Qwen-32B | 78.1 | 73.4 | 64.4 | 82.1 | 50.0 | 76.4 | 77.3 | 71.7 | 69.9 | 75.0 | 80.7 | 72.6 | 74.6 | 73.9 |
| R3-QWEN3-14B-LORA-4K | 56.3 | 66.0 | 62.7 | 79.5 | 41.7 | 79.8 | 65.3 | 64.5 | 64.1 | 56.3 | 74.8 | 68.4 | 65.9 | 67.2 |

**Poor Performance of GenRMs on PrincipleBench relative to Scalar RMs**   We find GenRMs generally under-perform Scalar RMs on PrincipleBench. The highest performing GenRM, Flexible Principles GenRM at 83.8, underperforms every ScalarRM. A hypothesis is that GenRMs are typically initialized from a reasoning model (Wang et al., 2025c), which are trained to excel on math, coding, and other logical reasoning benchmarks that only measure *correctness* of responses. Therefore, the initial reasoning process of such models over-indexes on response correctness (especially logical correctness in STEM fields) and less on other aspects such as absence of repetition and response clarity. This is also supported by the higher performance of Flexible Principles GenRM on JudgeBench and RM-Bench, which concern response correctness. This highlights the value of PrincipleBench to uncover previously under-addressed aspects of response quality beyond correctness.

# 5   ABLATION STUDIES

Table 4: Ablation of Scalar Reward Models on RM-Bench and JudgeBench. Higher is better.

| *Model* | Chat | Math | Code | Safety | Easy | Normal | RM-Bench Hard | Overall | Knowl. | Reason. | Math | Coding | JudgeBench Overall |
|---|---|---|---|---|---|---|---|---|---|---|---|---|---|
| Flexible Principles ScalarRM | | | | | | | | | | | | | |
| - Group Similarity=0.7 | 82.3 | 82.7 | 69.9 | 96.1 | 81.5 | 83.7 | 83.1 | 82.8 | 70.1 | 71.4 | 82.1 | 69.0 | 72.3 |
| - Group Similarity=0.8 (default) | 85.3 | 81.9 | 70.4 | 96.9 | 84.9 | 84.9 | 80.5 | **83.6** | 74.0 | 74.5 | 82.1 | 81.0 | **76.3** |
| - Group Similarity=0.9 | 82.9 | 77.7 | 70.9 | 95.9 | 82.2 | 82.7 | 80.7 | 81.9 | 70.8 | 73.5 | 85.7 | 69.0 | 73.7 |
| Flexible Principles ScalarRM | 85.3 | 81.9 | 70.4 | 96.9 | 85.5 | 84.9 | 80.5 | **83.6** | 74.0 | 74.5 | 82.1 | 81.0 | **76.3** |
| Fixed Principle Train Time | 74.2 | 82.4 | 70.4 | 92.7 | 85.3 | 82.3 | 72.2 | 79.9 | 69.5 | 66.3 | 82.1 | 76.2 | 71.4 |
| Fixed Principle Test Time | 84.6 | 80.7 | 69.9 | 92.2 | 85.3 | 84.3 | 79.6 | 81.9 | 70.1 | 69.4 | 76.8 | 69.1 | 70.9 |

**Group Similarity Threshold for Filtering Consensus Principles**   In Sec. 3, we chose 0.8 as the cosine similarity threshold for determining which principles to filter out. With a higher threshold, we increase the extent to which the principles across three annotators have to overlap *semantically* - which reduces the quantity of data post-filtering and removes principles that are due to the subjective preference of a single annotator. This threshold has *by far* the most impact on data quantity, inspiring an ablation study to use 0.7 or 0.9 as the threshold. The main dataset contains 33,000 samples with the default threshold of 0.8, which increases to 95,000 for threshold of 0.7 and reduces to 11,000 for threshold of 0.9. As shown in Tab. 4, the default threshold of 0.8 achieves the best RM-Bench and JudgeBench performance - indicating that the trade-off between data quantity and quality is optimal at this threshold.

**Fixing Principle to Accuracy of Information at Test Time**   We experimented with using the Flexible Principles ScalarRM but with Accuracy of Information fixed as the principle at Test Time. Unsurprisingly, this model performed substantially worse compared to the Flexible Principle ScalarRM. However, relative to a model that was only trained on a single fixed principle, the Fixed Principle Test Time model shows substantially better performance on RM-Bench (+2.0%) and similar performance on JudgeBench (-0.5%). This suggests that training a principle-following Reward Model can be beneficial even for users which only want to use a single principle at evaluation time. Such results echo previous findings that training on multiple tasks does not hurt and can sometimes boost performance on single tasks (Wei et al., 2022).

## 6 MODEL ALIGNMENT

Beyond evaluating the reward models' intrinsic performance, we also want to understand how they can be used to better align general-purpose LLMs. We conduct an alignment experiment with the Flexible Principles GenRM, the best performing model on RM-Bench and JudgeBench. A similar experiment with the Flexible Principles ScalarRM can be found in App. F.

**Evaluation**   Following Wang et al. (2025c), we use MT-Bench (Zheng et al., 2023), Arena Hard (Li et al., 2024) and WildBench (Lin et al., 2025) to evaluate our models. Notably, we use Arena Hard v2 that was recently released to replace Arena Hard v0.1, which was approaching saturation (>90% for top performing models). Arena Hard v2 uses a different set of prompts and a stronger baseline reference model (o3-mini-2025-01-31), meaning scores are generally much lower compared to the original and not directly comparable. Details in App. E.

**Training**   We conduct Reinforcement Learning training on top of the Qwen3-32B model (Yang et al., 2025b) using the GRPO (Group Relative Policy Optimization) algorithm (Shao et al., 2024) and the same dataset used to train the Flexible Principles GenRM. The actor/policy model generates multiple candidate responses without being explicitly aware of any principle, when given a conversation context that ends with a user question. The GenRM then evaluates the quality of these responses according to the specific judging principle associated with that training sample. Similar to Sec. 4.3, the actor is trained to optimize the objective (log-probs of Yes - log-probs of No) as rated by the GenRM, encouraging it to generate responses that conform as closely as possible to the principles. For training details, see App. C. For comparison, we use Bradley-Terry RM from Tab. 2 as a baseline RM for training the same policy model.

Table 5: Performance of Aligned Models. Higher is better for each metric except cost.

| Model | MT Bench (GPT-4-Turbo) | Arena Hard v2 (95% CI) | WildBench | | | | | | Cost | | |
| | | | Overall | Creative | Plan. | Data Analy. | Info. Seek. | Coding | In/M | Out/M | $ |
| --- | --- | --- | --- | --- | --- | --- | --- | --- | --- | --- | --- |
| Qwen3-32B | 9.38 | 44.0 (-1.6, +1.5) | 67.57 | 68.63 | 67.95 | 64.68 | 66.78 | 69.53 | 0.018 | 0.072 | **1x** |
| + RLBFF training | **9.50** | 55.6 (-1.6 / +1.4) | 70.33 | 71.73 | 70.73 | 69.37 | 68.96 | 70.94 | 0.018 | 0.072 | **1x** |
| + Baseline BT training | 9.45 | 47.5 (-1.4 / +1.7) | 67.38 | 68.42 | 68.13 | 65.32 | 66.34 | 68.49 | 0.018 | 0.072 | **1x** |
| *External Baselines* | | | | | | | | | | | |
| o3-mini | 9.26 | 50.0 (-0.0 / +0.0) | **71.64** | 69.04 | 72.44 | 74.37 | 65.81 | 73.21 | 1.1 | 4.4 | 61x |
| Claude-3.7-Sonnet (Thinking) | 8.93 | 54.2 (-2.0 / +1.8) | 65.45 | 66.72 | 65.94 | 63.59 | 63.08 | 67.36 | 3 | 15 | 188x |
| DeepSeek R1 | 9.49 | **57.4** (-2.0 / +2.0) | 64.24 | 70.75 | 66.29 | 59.20 | 68.56 | 61.04 | 0.4 | 2 | 25x |

**Results**   As shown in Tab. 5, our aligned model trained with the RLBFF technique using a fully open-source recipe and open-source data achieved similar or better performance across MT-Bench, Arena Hard v2 and WildBench, as compared to OpenAI o3-mini, Anthropic Claude-3.7-Sonnet (Thinking) and DeepSeek R1 (and substantially ahead of the Baseline BT-trained Qwen3-32B). Such performance is particularly impressive since our model is much cheaper to do inference with compared to R1, o3-mini and Claude-3.7-Sonnet. Based on OpenRouter (2025) in Sep 2025, Qwen3-32B only cost 1.8 cents per million input tokens and 7.2 cents per million output tokens, which is 24 to 187 times cheaper than compared models (assuming the same input/output tokens and a 1:1 ratio between input and output). Since our RLBFF-trained model uses an identical architecture as Qwen3-32B and can be served at the same cost, our RLBFF-trained model provides similar general-alignment capabilities compared to R1/o3-mini/Claude-3.7-Sonnet (Thinking) at a minuscule inference cost ($< 5\%$ of the cheapest alternative).

## 7 CONCLUSION

We propose Reinforcement Learning with Binary Flexible Feedback to combine the advantages of RLHF and RLVR. Leveraging RLBFF, we propose a recipe that utilizes open-source data to train reward models achieving state-of-the-art performance on RM-Bench, JudgeBench, and PrincipleBench. PrincipleBench, curated by us, is used to further evaluate the capacity of reward models to adhere to explicitly defined principles in reward assignment. Finally, we use RLBFF and our trained reward model to align Qwen3-32B to reach comparable performance as o3-mini and DeepSeek R1 on general alignment benchmarks of MT-Bench, WildBench and Arena Hard v2, while costing <5% for inference compared to those models.

## REPRODUCIBILITY STATEMENT

The procedure for pre-processing data has been described in Sec. 3. For experiments relating to Reward Modeling and Model Alignment, details are available in Sec. 4 and 6 as well as App. B, C, D, E, F, G and H.

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

## A   ELABORATIONS ON THEORETICAL BASIS FOR RLBFF APPROACH

The reward is derived as $\log P(Y) - \log P(N)$ (where $Y$ corresponds to the model generating "Yes", and $N$ generating "No"). As a result, such bias essentially only shifts the reward up or down. In practice we usually have $P(N) \simeq 1 - P(Y)$ (since the model is trained to only generate "Yes" or "No"), so the reward can be written $\log p - \log(1 - p)$ where $p = P(Y)$. Plotting this function of $p$ (e.g. in this link) shows that the shape of such a reward is roughly linear, except at the boundaries of large imbalance. This suggests that as long as the imbalance is not too severe, the induced bias corresponds approximately to a linear offset in the reward. In addition, we use GRPO for policy training, which normalizes the rewards in such a way that such a shift in reward is approximately being canceled out (up to the non-linearity of the reward function). In GRPO, the policy does not learn from examples where all generations are given the same reward (due to reward normalization). This does not happen with our reward definition since – except in very rare situations – different generations lead to different token probabilities in the Reward Model output, so there is always some training signal.

We also highlight a few interesting properties of our reward definition:

1. **It doesn't change the theoretical optimal policy** (when considering the reward alone, ignoring regularization): this optimal policy is by definition the one that always generates a response maximizing the expectation that the Reward Model generates "Yes" given the response to the prompt, which is equal to $P(Y) = p$. Since our reward can be written as $\log p - \log(1 - p)$, which increases monotonically with $p$, the optimality criterion is thus preserved.

2. Compared to using either a binary reward or $P(Y)$ directly, it puts more weight on the updates of responses that either fully satisfy the principle ($P(Y)$ close to 1) or completely fail at it ($P(Y)$ close to 0), which is a form of **"confidence-based" weighting**. This is because of the shape of $\log p - \log(1 - p)$ at the boundaries, where its derivative $\frac{1}{p(1-p)}$ takes steep values: deriving a mathematical justification is non-trivial as we also need to account for GRPO's reward normalization, but we provide an empirical validation of this intuition in a toy setting below (*).

3. Consider a situation where the policy only generates good responses for a given prompt, i.e. $p = P(Y)$ is distributed in some interval close to 1 across all responses, and assume that this distribution is such that $E[p]$ also the median (e.g., uniform or normal distribution). The strict convexity of the reward function $R(p) = \log p - \log(1 - p)$ near $p = 1$ implies $E[R(p)] > R(E[p])$ (Jensen's inequality), i.e. the mean reward is higher than the reward associated to the mean $p$ across all responses. A consequence is that on average more responses will get a reward $R(p)$ below $E[R(p)]$ than above it: this is because due to the previous assumption that $E[p]$ is the median of $p$'s distribution, 50% will get a reward below $R(E[p])$, and among those above some will fall somewhere between $R(E[p])$ and $E[R(p)]$. All those responses will be given a negative update in GRPO (due to the reward normalization that subtracts $E[R(p)]$ from all rewards), while the minority of best responses will be given a positive update. This encourages **exploitation** by focusing on the minority of very best responses when the model consistently outputs good generations. Conversely, if the policy only generates bad responses ($p$ close to 0), a similar reasoning relying on the concave shape of the reward function near 0 shows that most responses will be given a positive update, and the minority of worst responses a negative update. This encourages **exploration** by straying away from the minority of very worst responses and moving towards more diverse, better looking options. We believe such an adaptive exploitation / exploration mechanism may be beneficial to model training (compared e.g. to using binary rewards where there would be no training signal for rewards that are all 0 or all 1 for all responses, or using $R(p) = p$ which would result in balanced positive and negative updates in the settings previously described).

**Empirical validation of intuition using toy setting**   In addition, we describe the empirical simulation used to justify the theoretical benefits of the approach above. We average results over 10K runs, where in each run we sample 16 values of $p = P(Y)$ from a mixture of Gaussians: we first uniformly sample a number of modes $M$ between 1 and 5 (simulating diverse response qualities), and split the

(0, 1) interval into $M$ bins. Each value of $p$ is then sampled by first uniformly sampling a bin index, then sampling from a Gaussian centered within this bin. The standard deviation is set to the width of the bin, and any $p$ sampled outside of the chosen bin is re-sampled. We then compute the "relative weight" given to each sampled $p_i$ after GRPO reward normalization as $w(p_i) = \frac{|\tilde{R}(p_i)|}{\frac{1}{16} \sum_j |\tilde{R}(p_j)|}$, where $\tilde{R}(p_i)$ is the normalized GRPO reward: this gives us an indication of how influential that sample is in the GRPO update (whether negative or positive), compared to all samples. Finally, we report in the table below **the average weight across all simulations** as a function of $p$:

Table 6: Reward magnitude across probability intervals.

| Reward function $R(p)$ | $p \in (0, 0.01)$ | $p \in (0.01, 0.1)$ | $p \in (0.1, 0.9)$ | $p \in (0.9, 0.99)$ | $p \in (0.99, 1.0)$ |
|---|---|---|---|---|---|
| (ours) $\log p - \log(1-p)$ | 3.38 | 2.02 | 0.73 | 2.01 | 3.34 |
| $p$ | 1.88 | 1.70 | 0.83 | 1.69 | 1.86 |
| 1 with probability $p$, 0 otherwise | 1.01 | 1.00 | 1.00 | 1.00 | 1.01 |

We can see from Tab. 6 that **our reward definition gives more weight to samples close to the boundaries** ($p$ near 0 or 1), compared to either using $p$ directly as reward, or sampling a binary reward based on $p$.

## B  PROMPT TEMPLATES

**Extracting Principles and Fulfillment**
```
Feedback:  <feedback>
Generate a list of principles that the response is evaluated
against in the feedback.  For each principle, identify a
text span from the feedback relating to this principle and
then state whether the text span suggests that the response
satisfies the principle – yes/no/partially.  Return it as a json
dictionary in the format {"<principle 1>":  "<supporting text
span>-<yes/no/partially>", "<principle 2>":  "<supporting text
span>-<yes/no/partially>".}
```

**Scalar Reward Model**
```
<conversation>
Evaluate the response to the previous prompt in terms of whether
it satisfies this principle:  <principle>.  Only answer Yes or
No.
```

For evaluation, here are the principles used:

**PrincipleBench**

- Clarity: clarity of expression
- Accuracy: accuracy of facts
- Relevance: alignment with prompt
- No Repetition: avoidance of repetition
- Language Alignment: language compliance
- Essential Information: completeness of essential information
- Requirements Complete: adherence to prompt requirements

**RM-Bench**

- Math: correctness of answer
- Chat: accuracy of facts
- Code: accuracy

- Safety-Refuse: safety compliance
- Safety-Respond: compliance with prompt instructions

**JudgeBench**

- Knowledge: accuracy
- Reasoning: ethical compliance
- Math: correctness of answer
- Code: instruction compliance

**Generative Reward Model**
Prompt template:

> You are an expert evaluator tasked with assessing assistant responses based on specific principles. Below is a multi-turn conversation between a user and an assistant. Your task is to judge whether the last assistant response adheres to the specified principle.
> [Start of Conversation]
> User:
> xxxxx
> Assistant:
> xxxxx
> User:
> xxxxx
> Assistant:
> xxxxx
> [End of Conversation]
> [Start of Principle]
> xxxxx
> [End of Principle]
> Your task:
> 1. Carefully read the entire conversation.
> 2. Judge whether the assistant's final response adheres to the principle above.
> 3. Think step by step, citing concrete evidence from the conversation.
> 4. After your reasoning, output exactly: "Final Judgment: Yes" or "Final Judgment: No".
> Do NOT output anything after the line that contains the final judgment.

## C  TRAINING DETAILS

**Scalar Reward Model**    We train the Bradley-Terry model with 1 epoch of the HelpSteer3-Preference data as BT models are known to overfit beyond 1 epoch (Zhu et al., 2024; Wang et al., 2024). For fair comparison with BT models, all other Scalar Reward Models are also only trained with 1 epoch of our preprocessed dataset. For each model, we searched for the best Learning Rate among $\{1, 2, 3\}$e-6. We trained with the AdamW optimizer with 10 warm up steps and checkpoints saved every 50 steps. We use global batch size of 128 responses and a max sequence length of 4096. All Scalar Reward Models are trained with the NeMo-Aligner framework (Shen et al., 2024).

**Generative Reward Model**    We train Generative RMs for 3 epochs of our preprocessed data. For each rollout batch, there are 128 prompts and 8 responses per prompt. We use temperature=1.0 for generation and train the model with a batch size of 128 and a max sequence length of 8192. We use AdamW optimizer with 10 warm up steps and save checkpoints every 10 steps. For hyperparamters, we set learning rate as 2e-6 and search KL penalty among $\{1e-3, 1e-2, 1e-1, 2e-1, 3e-1\}$. The optimal KL penalty is 1e-2. During training and inference, we use the logprob returned by vLLM. Since vLLM only supports returning a maximum of 20 highest logprobs, if 'Yes' or 'No' are not in top-20, we set reward as -50. This both makes implementation easier in practice and ensures the model is better calibrated, through penalizing very low logprobs for either 'Yes' or 'No'. All Generative RM training is conducted with the NeMo-RL framework (NVIDIA, 2025b).

**Model Alignment** We conduct Reinforcement Learning training for 3 epochs of our preprocessed data. The actor generates 4 responses per prompt with temperature=1.0 and max sequence length 10240. We use AdamW optimizer with 10 warm up steps and save checkpoints every 10 steps. Each rollout batch contains 128 prompts, and each training batch contains 128 responses. For hyperparameters, we search over {1e-6, 2e-6, 3e-6} for learning rate and {0.01, 0.05} for KL penalty. The optimal KL penalty is 0.01. All alignment experiments are conducted with the NeMo-RL framework (NVIDIA, 2025b).

**Compute Requirements and Optimal Hyperparameters** are shown in Tab. 7

Table 7: Compute required and optimal hyperparameters for training each model, measured in H100-node-hours. Experiments are run on nodes of 8 H100-80GB SXM GPUs on internal clusters.

| *Model* | Compute (H100 node-hours) | LR | Step |
|---|---|---|---|
| **Scalar Reward Models** | | | |
| Bradley-Terry | 30 | 1e-6 | 350 |
| Fixed Principle | 30 | 1e-6 | 250 |
| Flexible Principles | 24 | 2e-6 | 256 |
| **Generative Reward Models** | | | |
| Flexible Principles | 512 | 2e-6 | 170 |
| **Aligned Models** | | | |
| RLBFF Actor | 864 | 2e-6 | 170 |

## D  REWARD MODEL RESULTS

Yu et al. (2025) reports the RM-BENCH performance of RewardAnything-8B-v1 at Overall 86.4. Across the domains, Chat is at 76.7, Math 90.3, Code 75.2 and Safety 90.2. Across difficulty, Easy is 89.4, Normal at 85.3 and Hard at 84.4.

THU-KEG (2025) found an issue with RewardAnything-8B-v1 that its reported overall score (86.4) is not equal to the average across Chat, Math, Code and Safety (76.7 + 90.3 +75.2 + 90.2) / 4 = 83.1.

To better understand how 86.4 was derived, we tested our hypothesis that this was calculated as a sample-level micro-average. Liu et al. (2024) states that RM-Bench contains 129 Chat, 529 Math, 228 Code and 441 Safety samples.

If we use the domain-average and calculate a sample-average, we get:

$$\frac{(76.7*129+90.3*529+75.2*228+90.2*441)}{(129+529+228+441)} = 86.4$$

Therefore, we conclude that RewardAnything-8B-v1's RM-Bench Overall Score was inaccurately calculated using the average across samples rather than the average across domain, as RM-Bench was intended to be (THU-KEG, 2025). To ensure fair comparison between RewardAnything-8B-v1 and all other models, we use the domain-level of 83.1 as its overall score and also derive the accurate difficulty level scores using a similar method.

## E  MODEL ALIGNMENT EVALUATION DETAILS

**Inference** Following Yang et al. (2025a), we performing inference on all models using vLLM with temperature 0.6, top-p 0.95 and max sequence length of 32,768. To ensure that the generation does not go beyond the max sequence length or causing out-of-memory, we set max generation length at 16,384. All models are evaluated with Thinking mode on.

**MT-Bench** Following Wang et al. (2025c); Meng et al. (2024); Wang et al. (2025a), we use GPT-4-0125-preview (GPT-4-Turbo) as a judge with human-verified reference answers for code, math

and reasoning categories. MT Bench contains 8 domains (Writing, Roleplay, Extraction, STEM, Humanities, Reasoning, Math and Coding) each with 10 tasks, containing two turns per task

**WildBench**  Following Wang et al. (2025c); Wake et al. (2025), we use WildBench score with the default GPT-4o-05-13 judge. WildBench contains 1024 diverse real-world prompts (variable-turns, not necessarily first turn) relating to Creative, Planning/Reasoning, Data Analysis/Math, Information/Advice seeking and Coding/Debugging.

**Arena Hard**  Arena Hard contains 500 challenging real-world, multilingual questions from Chatbot Arena (Zheng et al., 2023), with version 2 primarily relating to Coding and Math. For Arena Hard v2, we follow the official configuration to use the Gemini-2.5-Pro judge. We believe that Gemini-2.5-Pro judge is recommended over the GPT-4.1 judge because GPT-4.1 suffers from severe self-enhancement bias (Zheng et al., 2023), as it rates models developed by OpenAI substantially higher than other strong models. Specifically, LMSys (2024) shows that the top 7 models rated by GPT-4.1 are all from OpenAI and the next strongest model (Gemini-2.5-Pro) is rated as only slightly better than GPT-4.1-mini even though Gemini-2.5-Pro is widely considered as much stronger - ranking 1st vs 34th on Chatbot Arena on 10 Sep 2025 (Chiang et al., 2024). In contrast, Gemini-2.5-Pro is much less susceptible to such self-enhancement bias, rating 2 OpenAI models as stronger than itself (LMSys, 2024). In addition, we also do not use style control, as we found that the way it is calculated means that scores reported are not reproducible. Specifically, the official repository (LMSys, 2024) uses all locally-accessible model-judgments to calculate coefficients for features like markdown headings and response length. Therefore, the calculated score is highly dependent on the model judgments available locally, which can differ substantially across users. For instance, while the official leaderboard shows deepseek-r1 as 58.0 (-2.2 / +2.0) and QwQ-32B as 43.5 (-2.5 / +2.1), a local re-run of the recommended script without generating any new responses `python show_result.py --judge-names gemini-2.5 --control-features markdown length` shows deepseek-r1 as 50.1 (-1.8 / +2.1) and QwQ-32B as 35.6 (-1.6 / +1.8). Such large differences of around 8 points (or $> 3x$ of standard deviation for each model) makes it challenging to ensure style controlled scores are reproducible. On the other hand, non-style controlled scores remains consistent across various environments we tried at deepseek-r1 as 48.0 and QwQ-32B as 38.1.

## F  SCALARRM ALIGNMENT EXPERIMENT

Table 8: Performance of Aligned Models. Higher is better for each metric except cost.

| Model | MT Bench (GPT-4-Turbo) | Arena Hard v2 (95% CI) | WildBench | | | | | | Cost | | |
| | | | Overall | Creative | Plan. | Data Analy. | Info. Seek. | Coding | In/M | Out/M | $ |
|---|---|---|---|---|---|---|---|---|---|---|---|
| Llama-3.3-70B-Instruct (Init. Policy) | 8.29 | 5.7 (-0.8 / +0.7) | 52.5 | 55.5 | 54.1 | 48.2 | 54.8 | 51.7 | 0.10 | 0.32 | **1x** |
| + RLBFF training (ScalarRM) | **9.17** | 11.9 (-1.2 / +1.1) | 59.2 | 62.4 | 60.5 | 54.5 | 61.0 | 59.3 | 0.10 | 0.32 | **1x** |
| gpt-4o-2024-05-13 | 8.74 | **12.2** (-1.1 / +1.1) | **59.3** | 59.1 | 60.2 | 57.3 | 58.6 | 60.5 | 2.5 | 10 | 30x |

As shown in Tab. 8, an aligned model based on Llama-3.3-70B-Instruct with the ScalarRM is substantially worse compared to the model trained with the GenRM across MT Bench, Arena Hard, and WildBench. We believe this is because using the GenRM includes a reasoning process, which is conducive to aligning reasoning models (e.g., Qwen3-32B), which are substantially stronger in these benchmarks. Among the various benchmarks, the largest gap is seen in the Arena Hard V2 benchmark (11.9% vs. 55.6% of the Qwen3 RLBFF model in Table 5), which is likely because Arena Hard V2 contains a substantial amount of math and coding problems, which benefit from reasoning before response generation. Non-reasoning baseline models (Llama-3.3-70B-Instruct and gpt-4o) perform at a lower level on this benchmark.

We focused on GenRM for alignment setup because many of the popular general-domain benchmarks (MT Bench, Arena Hard, and WildBench) mainly focus on the correctness, even if they consider other aspects of the responses (e.g., stylistic suitability). Therefore, we prioritized RM-Bench and JudgeBench, which focus on the correctness of the response (at the reward model stage) as they better predict how well-aligned models perform. There are potentially certain domain-specific benchmarks that would align better with principles other than correctness (e.g., in character roleplay settings

where factual correctness matters less), but we want to focus on domain-general alignment and hence leave those as future work.

# G    SCALARRM EXPERIMENT WITH SMALLER BASE MODEL

Table 9: Performance of Small ScalarRMs on RM-Bench and JudgeBench. Higher is better.

| Model | RM-Bench | | | | | | | | JudgeBench | | | | |
| | Chat | Math | Code | Safety | Easy | Normal | Hard | Overall | Knowl. | Reason. | Math | Coding | Overall |
|---|---|---|---|---|---|---|---|---|---|---|---|---|---|
| Scalar RMs (<0.1 second/task) | | | | | | | | | | | | | |
| *Ours* | | | | | | | | | | | | | |
| Flexible Principles ScalarRM 8B | 75.0 | 65.1 | 58.0 | 91.9 | 80.2 | 73.7 | 63.7 | **72.5** | 64.3 | 63.3 | 69.6 | 57.1 | **64.0** |
| Bradley-Terry 8B | 72.5 | 64.2 | 55.5 | 92.3 | 82.9 | 73.9 | 56.6 | 71.4 | 58.4 | 63.3 | 80.4 | 54.8 | 62.9 |
| *External Baselines* | | | | | | | | | | | | | |
| Skywork-Reward-Llama-3.1-8B | 69.5 | 60.6 | 54.5 | 95.7 | 89.0 | 74.7 | 46.6 | 70.1 | 59.1 | 64.3 | 76.8 | 50.0 | 62.3 |

We observe in Tab. 9 that the Flexible Principles ScalarRM performs better compared to the Bradley-Terry RM trained on the same data, as well as an external baseline (Skywork-Reward-Llama-3.1-8B) across Overall RM-Bench and Overall JudgeBench.

# H    SCALARRM EXPERIMENT WITH BALANCED WEIGHTED DATA

Table 10: Performance comparison of equal-weighting of "Yes" and "No" samples across each principle on RM-Bench and JudgeBench. Higher is better.

| Model | RM-Bench | | | | | | | | JudgeBench | | | | |
| | Chat | Math | Code | Safety | Easy | Normal | Hard | Overall | Knowl. | Reason. | Math | Coding | Overall |
|---|---|---|---|---|---|---|---|---|---|---|---|---|---|
| *Ours* | | | | | | | | | | | | | |
| Flexible Principles ScalarRM | 85.3 | 81.9 | 70.4 | 96.9 | 85.5 | 84.9 | 80.5 | **83.6** | 74.0 | 74.5 | 82.1 | 81.0 | **76.3** |
| + Equal Weighting | 80.4 | 78.5 | 70.6 | 96.6 | 83.5 | 83.0 | 78.1 | 81.5 | 74.0 | 75.5 | 83.9 | 76.2 | **76.3** |

To better understand the impact of such imbalance in model performance, we perform reweighing to ensure that the proportion for Yes and No is equal across each principle. Specifically, for every principle with an imbalance, we randomly sub-sample the majority-class to match the minority-class. This results in retaining slightly under half the original data (with around 16 thousand samples). We find in Tab. 10 that ensuring equal weighing slightly reduced performance on RM-Bench and maintained the same performance on JudgeBench. This confirms our earlier hypothesis that the slight imbalance in distribution does not substantially impact performance.

