# OpenReview forum: "RLBFF: Binary Flexible Feedback to bridge between Human Feedback & Verifiable Rewards"
_ICLR.cc/2026/Conference — ICLR 2026 Poster_

### Official Review · Reviewer_Sidm · 2025-10-30

**Soundness:** 4
**Presentation:** 3
**Contribution:** 3
**Rating:** 6
**Confidence:** 2

**Summary:**

This paper introduces Reinforcement Learning with Binary Flexible Feedback (RLBFF), a novel paradigm designed to bridge the gap between RLHF and RLVR. RLBFF combines the versatility of human-driven preferences with the precision of rule-based verification. The method shows strong performance on RM-Bench and JudgeBench.

**Strengths:**

- The paper clearly articulates the distinct limitations of RLHF and RLVR. The proposed RLBFF is a well-motivated solution that effectively bridges these two paradigms by incorporating the flexibility of human judgment with explicit, binary, and verifiable criteria.

- Reward models trained with RLBFF achieve high performance on major alignment benchmarks.

**Weaknesses:**

The paper is technically sound and well-executed. I find no discernible weaknesses.

**Questions:**

N/A

---

> ### Author Response · Authors · 2025-11-21
> **Response to Reviewer Sidm**
>
> We thank the reviewer for their appreciation of our work. We encourage the reviewer to re-consider the assessment of our work given their comment in the Weakness section is limited to “The paper is technically sound and well-executed. I find no discernible weaknesses.” Is there anything we may clarify or expand upon in order to increase your confidence in the quality and relevance of our work?

---

### Official Review · Reviewer_Mh38 · 2025-10-31

**Soundness:** 3
**Presentation:** 4
**Contribution:** 3
**Rating:** 6
**Confidence:** 3

**Summary:**

Reinforcement Learning from Binary Flexible Feedback (RLBFF) proposes to use binary feedback derived from sets of explicit, human-interpretable principles to train scalar reward models and align large language models (LLMs). It mainly focuses on converting HelpSteer3-Feedback, an open-source dataset of natural language feedback, into structured binary evaluations along principles such as clarity, accuracy, or adherence to requirements. Unlike traditional RLHF, which relies on pairwise human preferences, or RLVR, which focuses narrowly on correctness, RLBFF extracts principle–fulfillment pairs (yes/no) to enable more transparent and fine-grained reward modeling. Using these data, the authors train flexible scalar and generative reward models that outperform previous methods on RM-Bench, JudgeBench, and the newly introduced PrincipleBench, achieving top leaderboard performance. They further apply RLBFF to align Qwen3-32B, achieving results comparable to proprietary systems like o3-mini and DeepSeek R1 on general alignment benchmarks such as MT-Bench, WildBench, and Arena Hard v2, while maintaining efficiency and interpretability at less than 5% of the inference cost.

**Strengths:**

* Merging RLHF and RLVR represents an important research direction, aiming to combine the interpretability and verifiability of rule-based feedback with the flexibility and broad applicability of human preference learning.
* RLBFF reports strong empirical results, where they outperform BT models trained on similar data and other Generative RMs on both in-distribution regarding training data (the proposed PrincipleBench) and other benchmarks(RM-bench, JudgeBench).
* The presentation is clear and easy to follow

**Weaknesses:**

* Extracting principles is not possible for other datasets that do not contain textual feedback. The format or quality of the textual feedback may be critical to this method's performance. This may limit the generalizability of this approach to other datasets.

**Questions:**

* The Skywork-8B-v2 model seems to report higher results on RM-bench, although on limited categories. Could you provide a comparison with your models if possible?
* In Table 5, can you include the BT baseline model in comparison with '+RLBFF'? This will help analyze whether the dataset is, or the reward model pipeline, is more critical for improving performance.

---

> ### Author Response · Authors · 2025-11-21
> **Response to Reviewer Mh38 (Part 1/2)**
>
> Thank you for the careful review!
>
> > Extracting principles is not possible for other datasets that do not contain textual feedback. The format or quality of the textual feedback may be critical to this method's performance. This may limit the generalizability of this approach to other datasets.
>
> We believe that textual feedback is a critical component for human-interpretable and robust data. For instance, when we shop online or decide on restaurants, seeing a 5-star rated product or restaurant without any users posting textual comments explaining their ratings may give the impression that the ratings could be fraudulent. Therefore, we see such dependence on textual-feedback as a feature of robust reward modelling rather than a limitation. We also note that this approach can be easily applied to other datasets, especially synthetically annotated ones since they commonly come with rationales behind the judgement. For example, popular datasets such as Ultrafeedback https://huggingface.co/datasets/openbmb/UltraFeedback or MT-Bench https://huggingface.co/spaces/lmsys/mt-bench come with textual feedback behind given ratings. The RLBFF method mainly seeks to effectively utilize such textual feedback from existing datasets, which we believe to provide a richer reward signal compared to the binary preference signal used in Bradley-Terry style modeling.
>
> > The Skywork-8B-v2 model seems to report higher results on RM-bench, although on limited categories. Could you provide a comparison with your models if possible?
>
> We do not believe that it’s fair to compare Skywork-8B-v2 models since they have been trained on 26 - 40 million preference pairs while our models have only been trained on 33 thousand samples, which is a thousand times smaller. Furthermore, our data has a transparent set of prompts and responses from open-source data while Skywork-v2 formula doesn’t explicitly state the sources of these prompts, causing some to hypothesize that it might be contaminated with test samples during the training process [1]  Therefore, we did not consider Skywork-v2 as a fair/useful baseline. In addition, the Skywork V2 was submitted to Arxiv on 1 July 2025, which is within 3 months of the ICLR submission deadline on 24 Sept 2025 and can be considered as contemporaneous work. Therefore, there is no obligation to compare against this model.
>
> However, we believe that Skywork-Reward-Llama-3.1-8B is a fair comparison given that it only utilizes 80 thousand preference-pairs that were released on Huggingface. To ensure fairness of comparison, we repeated our Flexible Principles ScalarRM recipe on the same initial model (Llama-3.1-8B-Instruct). We see that the Flexible Principles ScalarRM performs better compared to Skywork-Reward-Llama-3.1-8B as well as a Bradley-Terry model trained on the same data across Overall RM-Bench and Overall JudgeBench.
>
> |                                       | Chat  | Math  | Code  | Safety  | Easy  | Normal  | Hard  | **Overall RM-Bench**  | Knowl.  | Reason. |  Math  | Coding  | **Overall JudgeBench** |
> |---------------------------------------|-------|-------|-------|---------|-------|---------|-------|-----------------------|---------|---------|--------|---------|------------------------|
> | Flexible Principles ScalarRM 8B         | 75.0    | 65.1  | 58    | 91.9    | 80.2  | 73.7    | 63.7  | **72.5**                  | 64.3    | 63.3    | 69.6   | 57.1    | **64.0**                   |
> | Bradley-Terry  8B                       | 72.5  | 64.2  | 55.5  | 92.3    | 82.9  | 73.9    | 56.6  | 71.4                  | 58.4    | 63.3    | 80.4   | 54.8    | 62.9                   |
> | Baseline: Skywork-Reward-Llama-3.1-8B | 69.5  | 60.6  | 54.5  | 95.7    | 89.0    | 74.7    | 46.6  | 70.1                  | 59.1    | 64.3    | 76.8   | 50.0    | 62.3                   |

---

> ### Author Response · Authors · 2025-11-21
> **Response to Reviewer Mh38 (Part 2/2)**
>
> > In Table 5, can you include the BT baseline model in comparison with '+RLBFF'? This will help analyze whether the dataset is, or the reward model pipeline, is more critical for improving performance.
>
> We use the BT baseline model (from Table 2) to train a new policy model using the same hyperparameter as the +RLBFF training. Across MT Bench, Arena Hard V2 and WildBench, the BT baseline performs worse than the RLBFF training. This suggests that the RLBFF reward modelling pipeline is more critical to improve performance, as both the RLBFF reward model and the BT reward model are trained with the same data.
>
> | Model                                 | MT Bench | Arena Hard V2      | WildBench Overall | Creative | Plan. | Data Analy. | Info. Seek.  | Coding |
> |---------------------------------------|----------|--------------------|-------------------|----------|-------|-------------|--------------|--------|
> | Qwen3-32B                             | 9.38     | 44.0 (-1.6, +1.5)  | 67.57             | 68.63    | 67.95 | 64.68       | 66.78        | 69.53  |
> | + RLBFF training                     | 9.50     | 55.6 (-1.6 / +1.4) | 70.33             | 71.73    | 70.73 | 69.37       | 68.96        | 70.94  |
> | + BT baseline training             | 9.45     | 47.5 (-1.4 / +1.7) | 67.38             | 68.42    | 68.13 | 65.32       | 66.34        | 68.49  |
>
>
> [1] Zhilin Wang. Open-Data Reward Model Evaluations are Dead. 2025. URL: https://zhilin123.github.io/blog/2025/open/
>
> > Final Comments
>
> We hope our response has addressed your initial concerns and we would appreciate if you could consider raising your appraisal in light of it.

---

### Official Review · Reviewer_CRQz · 2025-11-01

**Soundness:** 3
**Presentation:** 3
**Contribution:** 3
**Rating:** 6
**Confidence:** 2

**Summary:**

This paper introduces Reinforcement Learning with Binary Flexible Feedback (RLBFF), aiming to bridge RL with human feedback (RLHF) and RL with verifiable rewards (RLVR) in the context of large language model alignment. The authors argue that RLHF suffers from ambiguity and reward hacking, whereas RLVR is too narrow, and propose RLBFF, which uses human-written, principle-grounded, binary feedback to make reward assignments both interpretable and precise. They extract over one thousand fine-grained principles from HelpSteer3-Feedback, train both scalar and generative reward models on this filtered data, and evaluate models on RM-Bench, JudgeBench, and the newly curated PrincipleBench. Their Scalar Reward Model supports user-specified principles at inference, runs efficiently, and achieves state-of-the-art results on several benchmarks. The authors also show how the approach can align Qwen3-32B to strong performance on major open-source alignment benchmarks with substantial cost savings.

**Strengths:**

1 Clear Motivation and Formulation: The paper gives a lucid account of the deficiencies of RLHF and RLVR, motivating the need for reward models that are both interpretable and principled (see Section 1, especially Table 1).

2 Methodological Innovation: The principle-grounded binary feedback format is an interesting compromise between free-form preferences and rigid verifiable criteria. The pipeline for extracting principles via LLM prompting, with grounding in supporting text spans and consensus filtering via embedding similarity, is thoughtfully designed (see Section 3 and the discussion around the filtering strategy).

3 Strong Empirical Results: The Flexible Principles reward models set a new high-water mark on RM-Bench and JudgeBench (see Table 2), and further excel on PrincipleBench, which focuses on adherence to explicit principles beyond correctness. This outperforms both scalar and generative baselines, including those requiring much heavier computation.

**Weaknesses:**

1 Limited Theoretical Analysis of Loss Design and Convergence: While the scalar reward modeling and binary prediction setup is elegant, there is little substantive mathematical analysis of the properties of the loss, especially with respect to class imbalance (Section 4.3). For example, while the principle fulfillment distribution is only slightly imbalanced ($35.4%$ no; $64.6%$ yes), no explicit techniques are described for dealing with such bias (e.g., reweighting, calibration). This omission is relevant since reward hacking and label bias are major themes of the work. More quantitative analysis of the loss landscape, as well as calibration studies, would add rigor.

2 Benchmarks and Biases: While Table 2 and Table 3 provide strong empirical evidence, most results are for benchmarks (RM-Bench, JudgeBench, PrincipleBench) that are closely related to the training data or evaluation principle. The generalizability to tasks or principles outside the extracted HelpSteer3-Feedback distribution is not explored in depth. Are there domains or principle types where the model underperforms or fails?

**Questions:**

Please see weaknesses.

---

> ### Author Response · Authors · 2025-11-21
> **Response to Reviewer CRQz (Part 1/3)**
>
> Thank you for the insightful suggestions!
>
> > Limited Theoretical Analysis of Loss Design and Convergence: While the scalar reward modeling and binary prediction setup is elegant, there is little substantive mathematical analysis of the properties of the loss, especially with respect to class imbalance (Section 4.3). For example, while the principle fulfillment distribution is only slightly imbalanced ( no;  yes) … More quantitative analysis of the loss landscape would add rigor.
>
> You are correct to point out that class imbalance may bias the reward model. However, as you mentioned, the distribution “is only slightly imbalanced”, so we do not expect it to be a problem in practice because:
>
> - The reward is derived as $\log P(Y) - \log P(N)$ (where $Y$ corresponds to the model generating "Yes", and $N$ generating "No"). As a result, such bias essentially only shifts the reward up or down. In practice we usually have $P(N) \simeq 1 - P(Y)$ (since the model is trained to only generate "Yes" or "No"), so the reward can be written $\log p - \log (1-p)$ where $p = P(Y)$. Plotting this function of $p$ (see e.g. https://www.desmos.com/calculator/n7ihbwhe3d) shows that the shape of such a reward is roughly linear, except at the boundaries of large imbalance. This suggests that as long as the imbalance is not too severe, the induced bias corresponds approximately to a linear offset in the reward.
> - We use GRPO for policy training, which normalizes the rewards such that a shift in reward is approximately canceled out (up to the non-linearity of the reward function).
> - In GRPO, the policy does not learn from examples where all generations are given the same reward (due to reward normalization). This does not happen with our reward definition since – except in very rare situations – different generations lead to different token probabilities in the Reward Model output, so there is always some training signal.
>
> We also highlight a few properties of our reward definition:
>
> 1. **It doesn’t change the theoretical optimal policy** (when considering the reward alone, ignoring regularization): this optimal policy is by definition the one that always generates a response maximizing the expectation that the Reward Model generates "Yes" given the response to the prompt, which is equal to $P(Y) = p$. Since our reward can be written as $\log p - \log (1-p)$, which increases monotonically with $p$, the optimality criterion is thus preserved.
> 2. Compared to using either a binary reward or $P(Y)$ directly, it puts more weight on the updates of responses that either fully satisfy the principle ($P(Y)$ close to 1) or completely fail at it ($P(Y)$ close to 0), which is a form of **“confidence-based” weighting**. This is because of the shape of $\log p - \log (1-p)$ at the boundaries, where its derivative $\frac{1}{p (1-p)}$ takes steep values: deriving a mathematical justification is non-trivial as we also need to account for GRPO's reward normalization, but we provide an empirical validation of this intuition in a toy setting below (*).
> 3. Consider a situation where the policy only generates good responses for a prompt, i.e. $p = P(Y)$ is distributed in some interval close to 1 across all responses, and assume that this distribution is such that $E[p]$ also the median (e.g., uniform or normal distribution). The strict convexity of the reward function $R(p) = \log p - \log(1-p)$ near $p = 1$ implies $E[R(p)] > R(E[p])$ (Jensen's inequality), i.e. the mean reward is higher than the reward associated to the mean $p$ across all responses. A consequence is that on average more responses will get a reward $R(p)$ below $E[R(p)]$ than above it: this is because due to the previous assumption that $E[p]$ is the median of $p$'s distribution, 50% will get a reward below $R(E[p])$, and among those above some will fall somewhere between $R(E[p])$ and $E[R(p)]$. All those responses will be given a negative update in GRPO (due to the reward normalization that subtracts $E[R(p)]$ from all rewards), while the minority of best responses will be given a positive update. This encourages **exploitation** by focusing on the minority of very best responses when the model consistently outputs good generations. Conversely, if the policy only generates bad responses ($p$ close to 0), a similar reasoning relying on the concave shape of the reward function near 0 shows that most responses will be given a positive update, and the minority of worst responses a negative update. This encourages **exploration** by straying away from the minority of very worst responses and moving towards more diverse, better looking options. We believe such an adaptive exploitation / exploration mechanism may be beneficial to model training (vs. using binary rewards where there would be no training signal for rewards that are all 0 or all 1 for all responses, or using $R(p) = p$ which would result in balanced positive and negative updates in the settings previously described).

---

> ### Author Response · Authors · 2025-11-21
> **Response to Reviewer CRQz (Part 2/3)**
>
> > no explicit techniques are described for dealing with such bias (e.g., reweighting, calibration). This omission is relevant since reward hacking and label bias are major themes of the work. .. calibration studies, would add rigor.
>
>
> **Reweighting experiment**
>
> To better understand the impact of such imbalance in model performance, we perform reweighing to ensure that the proportion for Yes and No is equal across each principle. Specifically, for every principle with an imbalance, we randomly sub-sample the majority-class to match the minority-class. This results in retaining slightly under half the original data (with around 16 thousand samples).
>
> |                               | Chat  | Math  | Code  | Safety  | Easy  | Normal  | Hard  | Overall RM-Bench  | Knowl.  | Reason. |  Math  | Coding  | Overall JudgeBench |
> |-------------------------------|-------|-------|-------|---------|-------|---------|-------|-----------------------|---------|---------|--------|---------|------------------------|
> | Flexible Principles ScalarRM  | 85.3  | 81.9  | 70.4  |  96.9   | 85.5  | 84.9    | 80.5  | 83.6                  | 74.0    | 74.5    | 82.1   | 81.0    | 76.3         |
> | \+ Equal Weighting             | 80.4 | 78.5  | 70.6  | 96.6    | 83.5  | 83      | 78.1  | 81.5                  | 74.0    | 75.5    | 83.9   | 76.2    | 76.3         |
>
> We find that ensuring equal weighing slightly reduced performance on RM-Bench and maintained the same performance on JudgeBench. This confirms our earlier hypothesis that the slight imbalance in distribution does not substantially impact performance.
>
>
> **Empirical validation of intuition using toy setting**
>
> (*) We describe here the empirical simulation used to justify point (2) above. We average results over 10K runs, where in each run we sample 16 values of $p = P(Y)$ from a mixture of Gaussians: we first uniformly sample a number of modes $M$ between 1 and 5 (simulating diverse response qualities), and split the (0, 1) interval into $M$ bins. Each value of $p$ is then sampled by first uniformly sampling a bin index, then sampling from a Gaussian centered within this bin. The standard deviation is set to the width of the bin, and any $p$ sampled outside of the chosen bin is re-sampled. We then compute the "relative weight" given to each sampled $p_i$ after GRPO reward normalization as $w(p_i) = \frac{|\tilde{R}(p_i)|}{\frac{1}{16} \sum_{j} |\tilde{R}(p_j)|}$, where $\tilde{R}(p_i)$ is the normalized GRPO reward: this gives us an indication of how influential that sample is in the GRPO update (whether negative or positive), compared to all samples. Finally, we report in the table below **the average weight across all simulations** as a function of $p$:
>
> | Reward function $R(p)$ | $p \in (0, 0.01)$ | $p \in (0.01, 0.1)$ | $p \in (0.1, 0.9)$ | $p \in (0.9, 0.99)$ | $p \in (0.99, 1.0)$ |
> | ----------------------------------- | ------- | ------- | ------- | ------- | ------- |
> | (ours) $\log p - log (1-p)$         | 3.38    | 2.02    | 0.73    | 2.01    | 3.34    |
> | $p$                                 | 1.88    | 1.70    | 0.83    | 1.69    | 1.86    |
> | 1 with probability $p$, 0 otherwise | 1.01    | 1.00    | 1.00    | 1.00    | 1.01    |
>
> We can see from this table that **our reward definition gives more weight to samples close to the boundaries** ($p$ near 0 or 1), compared to either using $p$ directly as reward, or sampling a binary reward based on $p$.
>
> **Discussion on Reward Hacking**
>
> Our method is in theory susceptible to reward hacking: it is not exactly the same as with RLVR where the verifier is typically an objective, deterministic function that only accepts valid responses. That being said, two key aspects of our method should contribute to making it overall **significantly more robust to reward hacking** compared to generic Reward Models:
>
> 1. Each principle focuses on a narrow, well-defined aspect of the response, so that “distractors” are less likely to impact the prediction; and
> 2. During training, each prediction task uses its own subset of principles (only a few out of \>1K), so that the policy is unlikely to see each principle often enough to be able to learn how to over-optimize it – especially considering that the policy is unaware of which principles are being applied by the Reward Model when evaluating a given task.

---

> ### Author Response · Authors · 2025-11-21
> **Response to Reviewer CRQz (Part 3/3)**
>
> > Benchmarks and Biases: While Table 2 and Table 3 provide strong empirical evidence, most results are for benchmarks (RM-Bench, JudgeBench, PrincipleBench) that are closely related to the training data or evaluation principle. The generalizability to tasks or principles outside the extracted HelpSteer3-Feedback distribution is not explored in depth. Are there domains or principle types where the model underperforms or fails?
>
> **We clarify that RM-Bench and JudgeBench benchmarks are not closely related to the training data**  (as they were curated externally of this work) and PrincipleBench does **not** use similar evaluation principles. Therefore, we believe that good performance on these benchmarks do indeed show generalization capabilities. For instance, RM-Bench contains a safety category, which is entirely-out-of-distribution from the training data, as it does not have feedback on any safety-relevant feature since HelpSteer3-Feedback curation filtered out all safety-related tasks [1]. Despite being out-of-distribution, the Flexible Principles RMs do well on the safety category scoring 95.5-96.9% accuracy, indicating good generalization to some unseen settings.
>
> On the other hand, Flexible Principles RMs do poorly on the Knowledge category of JudgeBench at 74.0 to 74.6%. The JudgeBench Knowledge category is constructed using model responses to MMLU-Pro prompts to college-level knowledge-intensive prompts across many disciplines such as health, psychology and chemistry. Each prompt has 4 to 10 possible options, of which only 1 is correct. We suspect that doing well in this category requires models to have advanced expertise in these subjects, which is not easily trained via reward modelling (and instead requires some form of specialized pre-training or retrieval-augmented generation).
>
> [1] Zhilin Wang, Jiaqi Zeng, Olivier Delalleau, Daniel Egert, Ellie Evans, Hoo-Chang Shin, Felipe Soares, Yi Dong, Oleksii Kuchaiev. 2025. HelpSteer3: Human-Annotated Feedback and Edit Data to Empower Inference-Time Scaling in Open-Ended General-Domain Tasks. ACL 2025.
>
> > Final Comments
>
> We hope our response has addressed your initial concerns and we would appreciate if you could consider raising your appraisal in light of it.

---

### Official Review · Reviewer_RKAF · 2025-11-01

**Soundness:** 3
**Presentation:** 3
**Contribution:** 3
**Rating:** 6
**Confidence:** 4

**Summary:**

The paper introduces Reinforcement Learning with Binary Flexible Feedback (RLBFF) that combines the versatility of RLHF with the precision of RLVR, enabling reward models (RMs) to capture aspects of response quality beyond correctness. RLBFF extracts principles that can be answered in a binary manner from natural language feedback. They create this training data with principles from Helpsteer3-Feedback, and use it to train reward models and also align LLMs. Experimentally, they demonstrate that RMs trained using principles outperform Bradley-Terry models and other external baselines on RM-Bench, JudgeBench, and a newly proposed PrincipleBench (designed to evaluate RMs ability to follow arbitrary principles). Additionally, they also show that a Qwen model aligned using RLBFF and their RM matches or exceeds the performance of o3-mini and Deepseek-R1 on general alignment benchmarks.

**Strengths:**

The paper proposes RLBFF, which combines the benefits of RLHF and RLVR, enabling reward models to capture arbitrary aspects of response quality beyond simple correctness. RLBFF extracts principles that can be answered in a binary fashion from feedback data and uses them to train both ScalarRMs and GenerativeRMs, improving their performance over standard Bradley–Terry models and external baselines on reward model benchmarks.

The proposed Flexible Principles Scalar RM is the first scalar reward model to enable grounding by user-specified principles, allowing for rapid inference while achieving enhanced performance over baselines.

The authors also perform alignment on a Qwen model using RLBFF and a trained principle GenRM, achieving performance that matches or exceeds o3-mini, Claude-Sonnet, and DeepSeek-R1 at a fraction of the inference cost.

Furthermore, the paper details the recipe used to construct the principle data from HelpSteer3-Feedback, improving reproducibility. Additionally, the authors propose PrincipleBench, a new reward model benchmark designed to evaluate the ability of RMs to follow arbitrary principles—a capability that was previously lacking.

**Weaknesses:**

The authors use Generative RMs in their alignment setup. However, inference with Generative RMs is more expensive than with Scalar RMs. A comparison against alignment performed using Scalar RMs would have been enlightening, as it would clarify the performance trade-off relative to the faster inference offered by Scalar RMs. Additionally, given the comparatively poorer performance of GenRMs compared to ScalarRMs on PrincipleBench, which was hypothesized to result from GenRMs being initialized from reasoning models that typically excel at correctness tasks, ScalarRMs might be a better choice for general purpose alignment.

It would also have been valuable to see whether RLBFF provides significant gains over approaches using standard Bradley–Terry RMs at smaller model scales.

**Questions:**

1) Could you provide alignment results obtained using a Scalar RM, and contrast its performance with that achieved using a Generative RM?

2) Given the comparatively poorer performance of GenRMs compared to ScalarRMs on PrincipleBench, which was hypothesized to result from GenRMs being initialized from reasoning models that typically excel at correctness tasks, wouldn’t ScalarRMs be a better choice for alignment in scenarios requiring broadly good performance, rather than just correctness?

3) Are there any results comparing RMs trained with RLBFF and Bradley–Terry models at smaller model scales (e.g., 3B, 8B)?

4) Will the principle dataset created from HelpSteer3-Feedback be released? While the recipe is provided, access to the dataset itself would greatly facilitate adoption and reproducibility.

---

> ### Author Response · Authors · 2025-11-21
> **Response to Reviewer RKAF (Part 1/2)**
>
> Thank you for your insightful review and suggestions:
>
> > The authors use Generative RMs in their alignment setup. However, inference with Generative RMs is more expensive than with Scalar RMs. A comparison against alignment performed using Scalar RMs would have been enlightening, as it would clarify the performance trade-off relative to the faster inference offered by Scalar RMs. Additionally, given the comparatively poorer performance of GenRMs compared to ScalarRMs on PrincipleBench, which was hypothesized to result from GenRMs being initialized from reasoning models that typically excel at correctness tasks, ScalarRMs might be a better choice for general purpose alignment.
>
> >Could you provide alignment results obtained using a Scalar RM, and contrast its performance with that achieved using a Generative RM?
>
> >Given the comparatively poorer performance of GenRMs compared to ScalarRMs on PrincipleBench, which was hypothesized to result from GenRMs being initialized from reasoning models that typically excel at correctness tasks, wouldn’t ScalarRMs be a better choice for alignment in scenarios requiring broadly good performance, rather than just correctness?
>
> Our earlier experiment in Section 6 performs alignment on a reasoning actor model (Qwen3-32B) with Generative RM, which reasons prior to giving a score. We believe that it is important to match reasoning/thinking mode between the actor model (i.e. initial policy model) and reward model. At the suggestion of the reviewer, we perform an additional alignment experiment with a non-reasoning actor model (Llama-3.3-70B-Instruct) with the non-reasoning Scalar RM.
>
>
> | Model                                 | MT Bench | Arena Hard V2      | WildBench Overall | Creative | Plan. | Data Analy. | Info. Seek.  | Coding |
> |---------------------------------------|----------|--------------------|-------------------|----------|-------|-------------|--------------|--------|
> | Llama-3.3-70B-Instruct (Init. Policy) | 8.29     | 5.7 (-0.8 / +0.7)  | 52.5              | 55.5     | 54.1  | 48.2        | 54.8         | 51.7   |
> | + RLBFF training (ScalarRM)            | 9.17     | 11.9 (-1.2 / +1.1) | 59.2              | 62.4     | 60.5  | 54.5        | 61.0         | 59.3   |
> | gpt-4o-2024-05-13                     | 8.74     | 12.2 (-1.1 / +1.1) | 59.3              | 59.1     | 60.2  | 57.3        | 58.6         | 60.5   |
>
> As shown in the table above, an aligned model based on Llama-3.3-70B-Instruct with the ScalarRM is substantially worse compared to the model trained with the GenRM across MT Bench, Arena Hard, and WildBench. We believe this is because using the GenRM includes a reasoning process, which is conducive to aligning reasoning models (e.g., Qwen3-32B), which are substantially stronger in these benchmarks. Among the various benchmarks, the largest gap is seen in the Arena Hard V2 benchmark (11.9% vs. 55.6% of the Qwen3 RLBFF model in Table 5), which is likely because Arena Hard V2 contains a substantial amount of math and coding problems, which benefit from reasoning before response generation. Non-reasoning baseline models (Llama-3.3-70B-Instruct and gpt-4o) perform at a lower level on this benchmark.
>
> We focused on GenRM for alignment setup because many of the popular general-domain benchmarks (MT Bench, Arena Hard, and WildBench) mainly focus on the correctness, even if they consider other aspects of the responses (e.g., stylistic suitability). Therefore, we prioritized RM-Bench and JudgeBench, which focus on the correctness of the response (at the reward model stage) as they better predict how well-aligned models perform. There are potentially certain domain-specific benchmarks that would align better with principles other than correctness (e.g., in character roleplay settings where factual correctness matters less), but we want to focus on domain-general alignment and hence leave those as future work.

---

> ### Author Response · Authors · 2025-11-21
> **Response to Reviewer RKAF (Part 2/2)**
>
> > It would also have been valuable to see whether RLBFF provides significant gains over approaches using standard Bradley–Terry RMs at smaller model scales.
>
> > Are there any results comparing RMs trained with RLBFF and Bradley–Terry models at smaller model scales (e.g., 3B, 8B)?
>
> We conducted additional reward modelling experiments at smaller scales using Llama-3.1-8B-Instruct as an initial model, and compare against Bradley Terry models with the same initial model
>
> |                                       | Chat  | Math  | Code  | Safety  | Easy  | Normal  | Hard  | **Overall RM-Bench**  | Knowl.  | Reason. |  Math  | Coding  | **Overall JudgeBench** |
> |---------------------------------------|-------|-------|-------|---------|-------|---------|-------|-----------------------|---------|---------|--------|---------|------------------------|
> | Flexible Principles ScalarRM 8B              | 75.0 | 65.1  | 58.0  | 91.9    | 80.2  | 73.7  | 63.7  | **72.5**  | 64.3    | 63.3    | 69.6   | 57.1    | **64.0**  |
> | Bradley-Terry  8B                                      | 72.5  | 64.2  | 55.5  | 92.3    | 82.9  | 73.9  | 56.6  | 71.4                  | 58.4    | 63.3    | 80.4   | 54.8    | 62.9                   |
> | Baseline: Skywork-Reward-Llama-3.1-8B | 69.5  | 60.6  | 54.5  | 95.7    | 89.0 | 74.7    | 46.6  | 70.1                  | 59.1    | 64.3    | 76.8   | 50.0    | 62.3                   |
>
> We observe that the Flexible Principles ScalarRM performs better compared to the Bradley-Terry RM trained on the same data, as well as an external baseline (Skywork-Reward-Llama-3.1-8B) across Overall RM-Bench and Overall JudgeBench.
>
> > Will the principle dataset created from HelpSteer3-Feedback be released? While the recipe is provided, access to the dataset itself would greatly facilitate adoption and reproducibility.
>
> The principle dataset has been released on Huggingface alongside the model weights of the reward and aligned models, which have been downloaded more than a few thousand times by the community.
>
> > Final Comments
>
> We hope our response has addressed your initial concerns and we would appreciate if you could consider raising your appraisal in light of it.

---

### Comment · Area_Chair_mKJC · 2025-11-25

Dear reviewers:

The authors have submitted their rebuttal, and we now require your follow-up assessments to move the decision process forward. Please review the authors’ responses and update your evaluations accordingly.

Your prompt follow-up is necessary for us to finalize the meta-review.

Kindly submit your updates as soon as possible.

Best,

Area Chair

---

### Author Response · Authors · 2025-12-02
**General Comment for Area Chair**

We thank the reviewers for their kind recognition of our contributions and helpful feedback. Specifically, reviewers found our work to be well-motivated (reviewer Sidm, CRQz), innovative (reviewer CRQz) and important (reviewer Mh38). In addition, all reviewers (Sidm, CRQz, Mh38, RKAF) appreciate the strong empirical results demonstrated in reward and aligned model training. As some reviewers recommended follow-up experiments (e.g. training with different model sizes, alternative alignment setups and data preprocessing procedures) or had clarification questions, we carried out all recommended experiments and gave detailed responses to their questions. We also show that these new results are in line with our initial findings. While the reviewers did not have the opportunity to revise their initial scores based on our author-response due to the Nov 27 incident, we hope that the area chair can consider the unanimous acceptance recommendation made by all four reviewers in making the final recommendation for this paper.

---

### Meta-Review · Area_Chair_6Fw5 · 2026-01-06

**Summary:**

This is quite an interesting paper, and all the reviewers have agreed to accept it.

**Reviewer Scores:**

None

---

### Decision · Program_Chairs · 2026-01-26

Accept (Poster)